# Predicting 5-year survival after kidney transplantation in Colombia using the survival benefit estimator tool

**Laura Nino-Torres[1], Andrea García-Lopez[2], Nasly Patino-Jaramillo[2], Fernando Giron-Luque[1], Alejandro Nino-Murcia[1]***

**1** Department of Transplant Surgery, Colombiana de Trasplantes, Bogotá, Colombia, **2** Department of Transplant Research, Colombiana de Trasplantes, Bogotá, Colombia

\* anino@colombianadetrasplantes.com

## Abstract

### Introduction

A complex relationship between donor and recipient characteristics influences kidney transplant (KT) success. A tool developed by Bae S. et al. (Survival Benefit Estimator, SBE) helps estimate post-KT survival. We aim to evaluate the predictive performance of the SBE tool in terms of 5-year patient survival after a kidney transplant.

### Methods

A retrospective cohort study of all deceased-donor KT recipients between January 2009 to December 2021. A descriptive analysis of clinical and sociodemographic characteristics was performed. The SBE online tool was used to calculate the predicted patient survival (PPS) and the survival benefit at five years post-KT. Comparisons between predictive vs. actual patient survival were made using quintile subgroups. Three Cox regression models were built using PPS, EPTS, and KDPI.

### Results

A total of 1145 recipients were evaluated. Mortality occurred in 157 patients. Patient survival was 86.2%. Predictive survival for patients if they remained on the waiting list was 70.6%. The PPS was 89.3%, which results in a survival benefit (SB) of 18.7% for our population. Actual survival rates were lower than the predicted ones across all the quintiles. In unadjusted analysis, PPS was a significant protective factor for mortality (HR 0.66), whereas EPTS (HR 8.9) and KDPI (HR 3.25) scores were significant risk factors. The discrimination of KDPI, PPS, and EPTS scores models were 0.59, 0.65, and 0.66, respectively.

### Conclusion

SBE score overestimated actual survival rates in our sample. The discrimination power of the score was moderate, although the utility of this tool may be limited in this specific population.

**Data Availability Statement:** Data cannot be shared publicly because of center regulations. Data may be available from the Colombiana de Trasplantes / Ethics Committee (Dexa Diab) for

researchers who meet the criteria for access to confidential data.

**Funding:** The authors received no specific funding for this work.

**Competing interests:** The authors have declared that no competing interests exist.

## Introduction

Kidney transplantation (KT) is the gold standard for end-stage kidney disease. Given organ shortages and limited resources, it is important to focus on improving graft and patient survival [1]. The kidney allocation system (KAS) is key in the process of organ procurement, organ supply, and transplantation [2]. The KAS should efficiently prevent higher deceased donor kidney discard rates [3]. A complex relationship between donor and recipient characteristics influences post-transplant success. The United States Organ Procurement and Transplantation Network (OPTN) based its KAS on the Estimated post-transplant index (EPTS) and the Kidney donor profile index (KDPI).

The EPTS is a numerical measure to allocate kidneys in the KAS. The Kidney donor profile index (KDPI) [4, 5] estimates the risk of post-transplant graft failure based on cadaveric donor characteristics. A lower percentage in both scores (EPTS and KDPI) suggests the highest benefit in kidney transplants [6]. Notwithstanding, the literature showed lower mortality risk in the recipients with kidney transplants at any rate of EPTS/KDPI score compared to patients who remained on the waiting-list and dialysis [7]. That being the case, a kidney transplant is truly worth performing for the potential recipients. The Survival Benefit Estimator (SBE) is a tool developed by Bae S. et al. to help estimate post-transplant survival based on donor and recipient characteristics using the KDPI and EPTs scores [7, 8].

Some studies published the application of the EPTS and KDPI in non-US countries [3, 6, 9]. The prospect of the use of EPTS/KDPI scores in another allocation international systems faces difficulties, one of which is the difference between the transplant programs, allocation system design, donor and recipient characteristics [6, 10, 11].

In Colombia, our allocation system does not apply EPTS or KDPI for national kidney assignments. Nevertheless, these clinical tools may help to optimize the selection of donor-recipient pairs and individualize decision-making for clinical kidney offers. Therefore, we aim to evaluate the predictive performance of the SBE tool in terms of 5-year patient survival after a kidney transplant in a Colombian cohort.

## Methods

### Study design and population

A retrospective cohort of deceased kidney transplant recipients transplanted in Colombiana de Trasplantes was analyzed. Colombiana de Trasplantes is a transplant network in Colombia with four centers that perform around 14.5% of the annual national kidney transplant activity. We included recipients of deceased donors aged ≥ 18 years transplanted between January 1, 2009, to December 31, 2021. We excluded patients with insufficient information to calculate the KDPI and ETPS score (age, height, weight, ethnicity/race, history of hypertension, history of diabetes, cause of donor death, serum creatinine, hepatitis C virus status, and whether the donor met criteria for circulatory death; recipient age, diabetes, time on dialysis and previous solid organ transplant) (Fig 1). Recipients were followed up to graft failure, death, or end of follow up at 5 years post-transplantation, whichever was earliest. Donor and recipient variables were collected from institutional medical records; when patients missed appointments, we called them and their families and reviewed the ADRES database to ensure their alive status.

### Immunosuppression and follow-up protocol

Maintenance immunosuppression protocol includes a dual therapy with calcineurin inhibitors-based therapy and mycophenolate mofetil, steroids-free. Patients are monitored closely in the first four weeks post-transplantation, and they return for follow-up monthly thereafter.

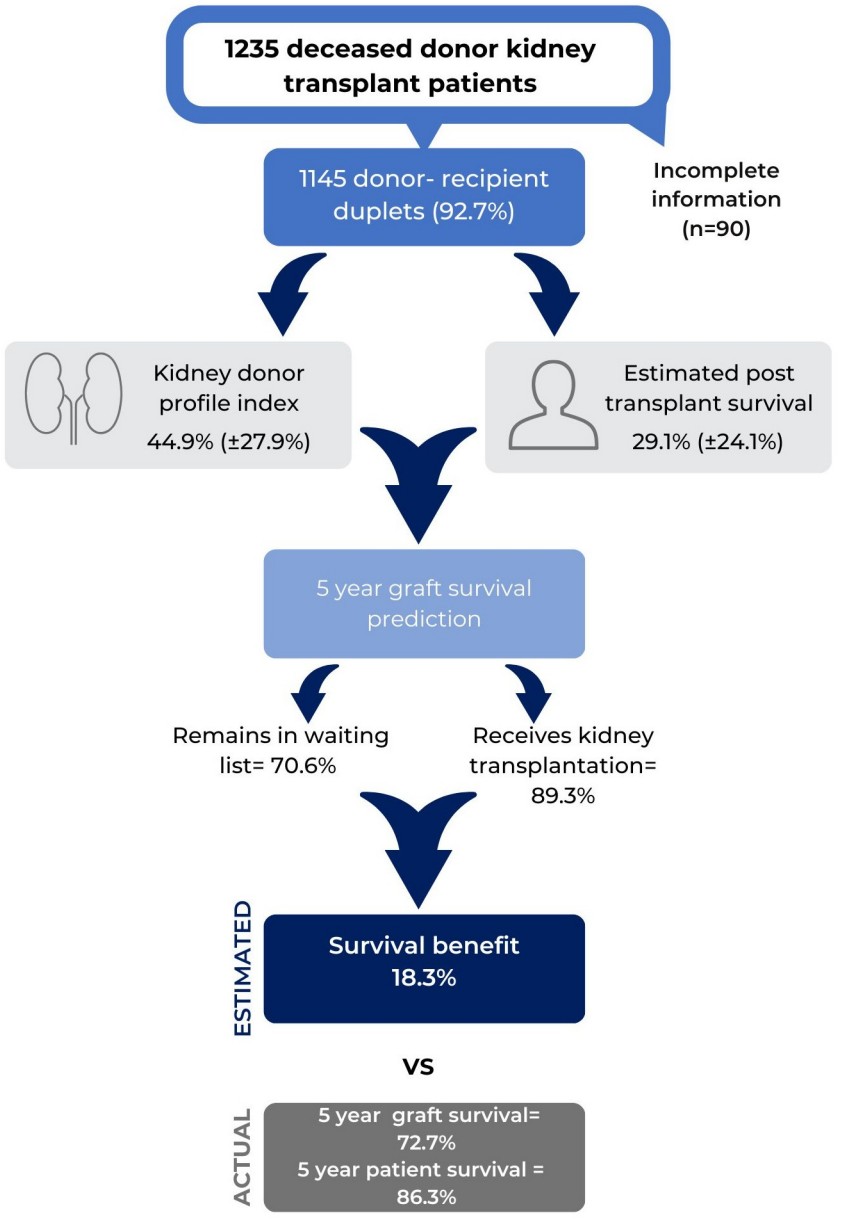

**Fig 1. Methods and results graphic representation.**

## Kidney donor allocation

Organ allocation is made according to the allocation criteria for kidney transplantation in Colombia [12]. The National Health Institute defines these criteria and applies only to organs of deceased brain donors (DBD). They consider the geographical location within the nation, blood type, accumulated time on the waiting list, HLA mismatch, sensitization, and loss of vascular access [13, 14]. KDPI / KDRI indices are not considered for taking or allocating organs [10]. KDPI and EPTS online calculators were used for research purposes, not organ allocation [14].

## Outcomes

**Survival estimation and actual survival.** The main outcome was the predicted patient survival (PPS) at 5 years post-KT. The secondary outcome was the survival benefit (SB), defined as the absolute reduction in mortality risk with KT compared with remaining on dialysis expressed in percentage points. The PPS and SB were calculated using the prediction online tool: KDPI-EPTS Survival Benefit Estimator developed by Bae et al. [7] available at www. transplantmodels.com/kdpi-epts/. Briefly, the PPS and SB at 5 years post-KT were estimated based on a combination of donor quality and candidate condition measured using the KDPI [15] and the EPTS [16] scores as metrics of donor and candidate risk.

To define actual post-transplant patient survival, we used the definition proposed by Coca et al. [6] as the time from KT to death, censoring by the end of the study. All patients were followed-up long enough to establish 60-month survival, regardless of the allograft status (functioning or not).

To compare the predicted vs. actual post-transplant patient survival, we stratified our sample by PPS quintiles. Unadjusted and adjusted models were built using PPS, EPTS, and KDPI.

**Statistical analysis.** Descriptive statistics were used to report population characteristics. Data was expressed as mean and standard deviation (SD) or median and range for non-normal distributions. Qualitative variables were described as frequencies and percentages. There were no missing values. Overall actual survival probability was estimated using the Kaplan–Meier method and compared by the log-rank test. We use quintile distribution to obtain five groups of comparable size to compare the PPS with our actual patient survival. We used an unadjusted and adjusted Cox proportional hazard regression to obtain Hazard ratios (HRs) with 95% confidence interval (CI) of three models (i) PPS, (ii) KDPI, and (iii) EPTS. Adjusted variables were selected by clinical and statistical significance in the bivariate analysis and were excluded the variables that are considered in each score (PPS, KDPI, EPTS). Cox Model discrimination was estimated using Harrell´s C index, where 0.5 means no discrimination, and 1 means perfect discrimination. A p-value $<0.05$ was accepted as statistically significant. Statistical analyses were performed in R 3.4.2.

## Ethics considerations

This study was approved by the ethics research committee of the institution, acting in concordance with local and national regulations, as well as with the Helsinki Declaration [17, 18]. Since this retrospective cohort was considered a non-risk study by the Colombian Resolution 8430 of 1993 [19], the ethics committee waived the requirement of informed consent.

## Results

### Recipient characteristics

A total of 1250 KT were performed between January 2009 and December 2021. The final analysis included 1145 recipients and 814 deceased donors. The median age of recipients was 47 years (IQR 37–56), approximately 60% were male (n = 693), 17.1% had a history of diabetes, and the most frequent dialysis type was hemodialysis (n = 667;58.3%). The main known cause of CKD was glomerular (n = 209;18.3%). Recipient characteristics are described in Table 1.

**Donor characteristics.** Donors' median age was 44 years (IQR 29–54), approximately 60% were male (n = 493), median BMI was 24.7 m$^2$/kg (IQR 22.9–27.2), 21.1% had a history of hypertension, 2.6% had history of diabetes, main cause of death was cerebrovascular/stroke (n = 394; 48.4%) and median serum creatinine was 0.9 mg/dl (IQR 0.6–1.1) (Table 2).

**Table 1. Characteristics of kidney transplant recipients.**

| Recipient characteristics | Total |
|---|---|
| | (N = 1145) |
| **Sex, n (%)** | |
| Female | 452 (39.5) |
| Male | 693 (60.5) |
| **Age years, median [IQR]** | 47 [37, 56] |
| **Diabetes, n (%)** | 196 (17.1) |
| **Dialysis time months, median [IQR]** | 30 [10, 68] |
| **Dialysis type, n (%)** | |
| Hemodialysis | 667 (58.3) |
| Peritoneal | 430 (37.6) |
| Predialysis (preemptive transplant) | 48 (4.2) |
| **Number of previous organ transplants, n (%)** | |
| 0 | 1057 (92.3) |
| 1 | 77 (6.7) |
| 2 | 11 (1.0) |
| **EPTS, median [IQR]** | 21% [10, 43] |
| **EPTS, n (%)** | |
| <20 | 560 (48.9) |
| 20–80 | 529 (46.2) |
| >80 | 56 (4.9) |
| **Underlying cause of CKD, n (%)** | |
| Unknown | 403 (35.2) |
| Glomerular | 209 (18.3) |
| Hypertensive | 184 (16.1) |
| Diabetic | 174 (15.2) |
| Congenital | 85 (7.4) |
| Other | 50 (4.4) |
| Obstructive | 39 (3.4) |

Abbreviations: IQR: Interquartile range; EPTS: Estimated Post Transplant Survival; CKD: chronic kidney disease

**Distribution of KDPI and ETPS scores.** Of the total of recipients, 560 had an EPTS score of 0–20, 529 had an EPTS of 20–80, and a low number of recipients had a high EPTS score (n = 56; 4.8%). Most donors presented KDPI of 20–80%. The distribution of KDPI and EPTS scores in our cohort is shown in Fig 2.

## Actual patient survival and predicted patient survival

During the follow-up, mortality occurred in 157 patients. At 5 years, actual patient survival was 86.2% (CI 95%, 84.3%-88.3%). Predictive survival for patients if they were to remain wait-listed for 5 years was 70.6%, and if they were to receive the transplant (PPS) was 89.3%, which results in a survival benefit (SB) of 18.7% for our population (Table 3). Furthermore, there was a higher SB for patients older than 65 years (24%).

The study sample was divided into quintiles using the results of the PPS. The comparison of the average predicted survival rate with the actual survival in each PPS quintile is shown in Fig 3. Actual survival rates were lower than the predicted ones across all the quintiles. In unadjusted analysis, a higher predicted survival rate was a significant protective factor for recipient

**Table 2. Donor characteristics.**

| Donor characteristics | Total |
|---|---|
| | (N = 814) |
| **Donor age in years, median [IQR]** | 44 [29, 54] |
| **Donor sex, n (%)** | |
| Female | 321 (39.4) |
| Male | 493 (60.6) |
| **BMI, median [IQR]** | 24.7 [22.9, 27.2] |
| **History of Hypertension, n (%)** | |
| Unknown | 13 (1.6) |
| No | 629 (77.3) |
| yes, 0–5 years | 53 (6.5) |
| yes, 6–10 years | 28 (3.4) |
| yes, > 10 years | 46 (5.7) |
| yes, unknown duration | 45 (5.5) |
| **History of Diabetes, n (%)** | |
| Unknown | 8 (1.0) |
| No | 785 (96.4) |
| yes, 0–5 years | 8 (1.0) |
| yes, > 10 years | 3 (0.4) |
| yes, unknown duration | 10 (1.2) |
| **Cause of Death, n (%)** | |
| Cerebrovascular/stroke | 394 (48.4) |
| Head trauma | 365 (44.8) |
| Anoxia | 30 (3.7) |
| SNC tumor | 15 (1.8) |
| Other | 10 (1.2) |
| **Serum Creatinine (mg/dL), median [RIC]** | 0.9 [0.69, 1.16] |
| **KDPI, median [RIC]** | 42% [19, 66.7] |
| **KDPI categories, n (%)** | |
| **<20** | 218 (26.8%) |
| **20–80** | 492 (60.4%) |
| **>80** | 104 (12.8%) |

Abbreviations: IQR: Interquartile range; BMI: body mass index; KDPI: Kidney Donor Profile Index; CNS: central nervous system

death (HR 0.66; CI 95% 0.58–0.75), whereas EPTS (HR 8.9; CI 95% 5.4–15.4) and KDPI (HR 3.25; CI 95% 1.8–5.7) scores were significant risk factors for recipient death. HR for KDPI was adjusted with the recipient's age, sex, diabetes history, months in dialysis, and CKD etiology. The EPTS model was adjusted by donors' age, sex, BMI, death cause, diabetes, and hypertension history. Finally, the predicted survival model was adjusted by recipients' sex, diabetes history, months in dialysis, CKD etiology, donors' sex, BMI, and cause of death. The discrimination of the KDPI model was low, with a Harrell´s C of 0.59 compared with predicted survival and EPTS score models, with a Harrell´s C of 0.65 and 0.66, respectively (Table 4).

## Actual patient survival according to EPTS and KDPI scores

Patient survival at 5 years post-KT was 92.5% (CI 95%, 90.3–94.7) for recipients with an EPTS score of 0–20, 82.2% (CI 95%, 79.0–85.6) with an EPTS of 20–80 and 62.5% (CI 95%, 51.0–

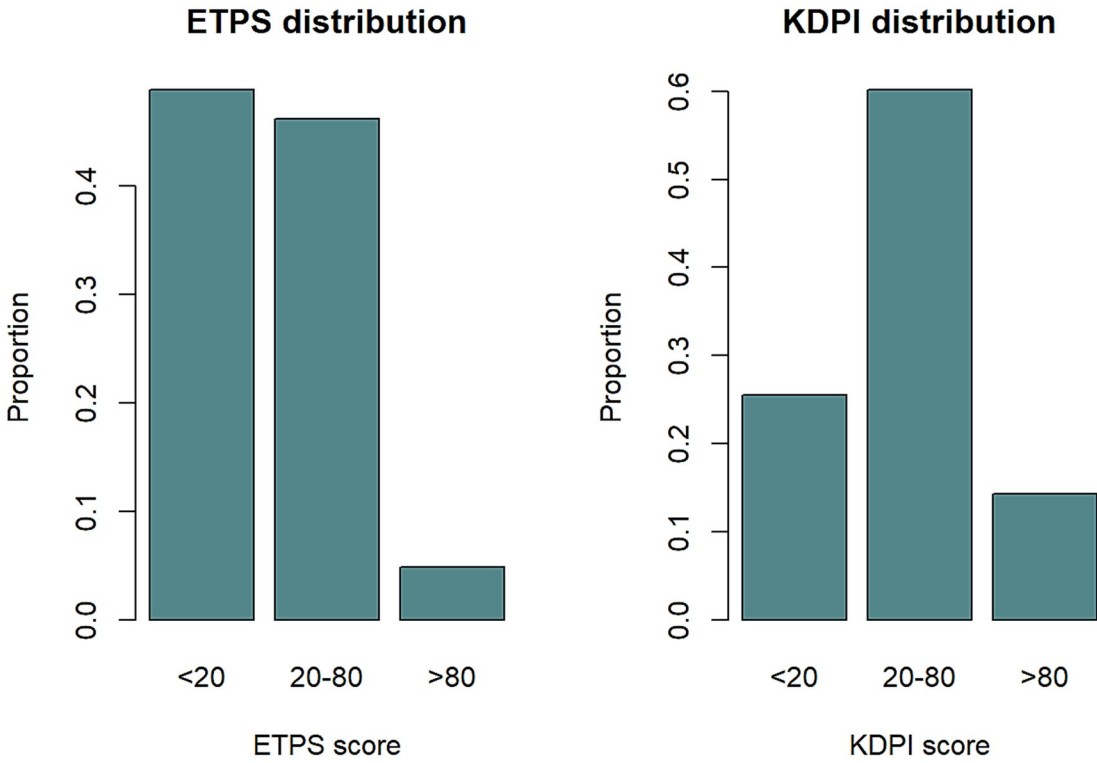

**Fig 2. EPTS and KDPI distribution adjusted by age groups.**

76.6) with an EPTS of 80–100 (p-value <0.001). For recipients with a KDPI score of 0–20, patient survival at 5 years post-KT was 89.4% (CI 95%, 85.9–93), with a KDPI of 20–80 was 86.6% (84.1–89.2), and with a KDPI of 80–100 was 79.3% (CI 95%, 73.3–85.7) (p-value <0.005), Fig 4.

## Discussion

A complex relationship between donor and recipient characteristics influences post-transplant success. Therefore, clinical tools may help to optimize the selection of donor-recipient pairs and individualize decision-making for clinical kidney offers. The tool developed by Bae S. et al. [7] helps to estimate post-transplant survival based on donor and recipient conditions. The present study aimed to evaluate the predictive performance of the SBE tool in terms of 5-year patient survival after a kidney transplant in a Colombian cohort.

**Table 3. Main outcomes at 5-years post-transplantation based on the KDPI-EPTS survival benefit estimator.**

| Outcomes | |
|---|---|
| **Predictive waiting list survival,** mean (SD) | 70.6% (12.7) |
| **Predictive kidney transplant survival,** mean (SD) | 89.3% (8.0) |
| **Predictive overall survival benefit,** mean (SD) | 18.7% (6.4) |
| **Predicted survival benefit according to age groups**, mean (SD) | |
| 18–65 | 18.4% (6.4) |
| >65 | 24% (4.2) |
| **Actual patient survival** | 86.2% (84.3–88.3) |

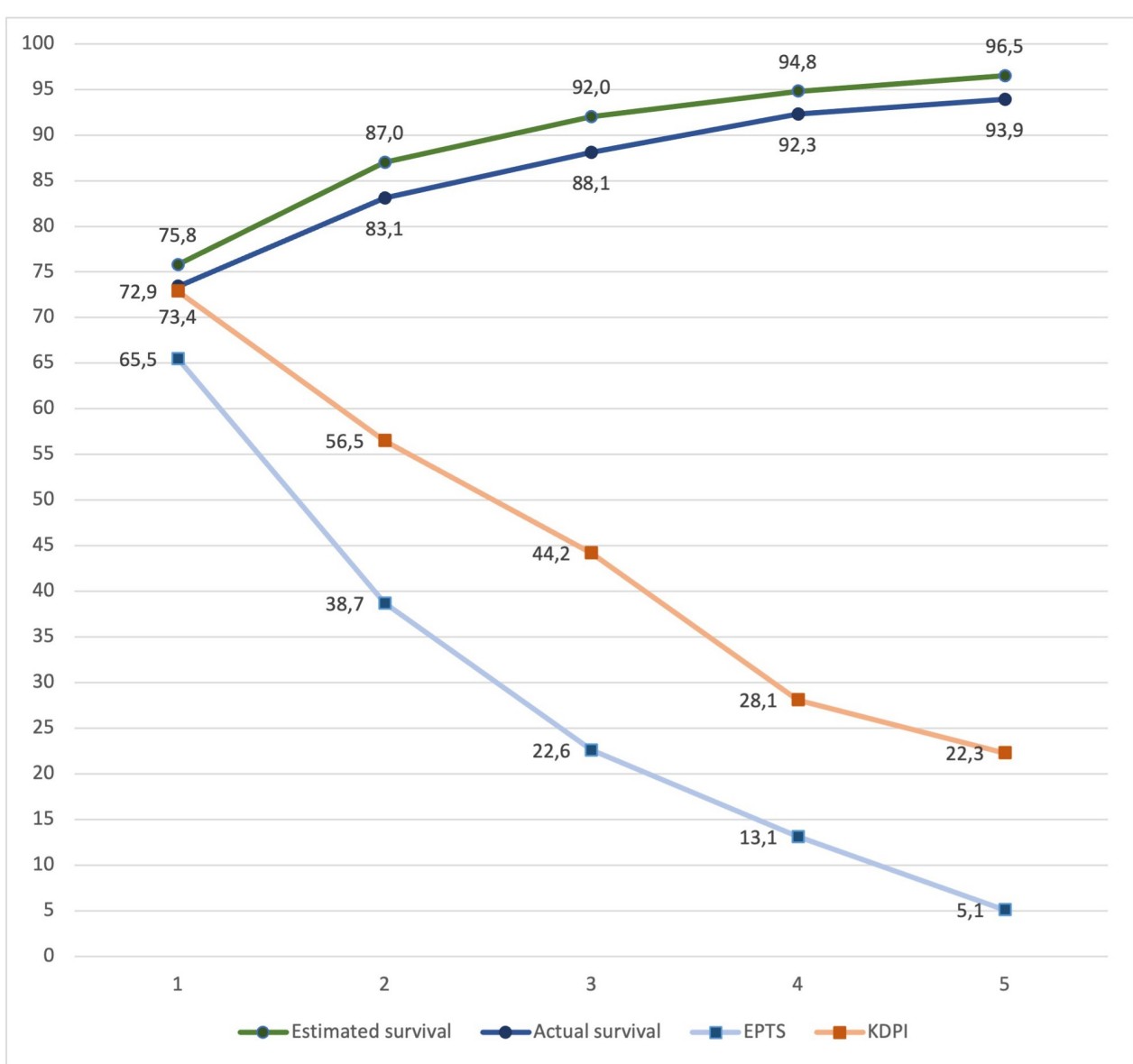

**Fig 3. Comparison of estimated survival calculated by PPS and the actual survival.**

**Table 4. Association between predicted survival, EPTS, KDPI scores and recipient death by Cox regression modeling.**

|  | Unadjusted HR | 95% CI | P value | C-index |
|---|---|---|---|---|
| Predicted survival | 0,66 | 0.58–0.75 | <0.000 | 0,65 |
| KDPI | 3,25 | 1.85–5.75 | <0.001 | 0,59 |
| EPTS | 8,9 | 5.14–15.41 | <0.001 | 0,66 |
|  | **Adjusted HR** | **95% CI** | **P value** | **C-index** |
| Predicted survival | 0.003 | 0.00–0.031 | <0.000 | 0,69 |
| KDPI | 1,67 | 0.87–3.197 | 0.12 | 0.69 |
| EPTS | 8,16 | 4.35–15.2 | <0.001 | 0,68 |

CI confidence interval, PPS predictive patient survival, EPTS estimate post-transplant survival, HR hazard ratio, KDPI kidney donor profile index.

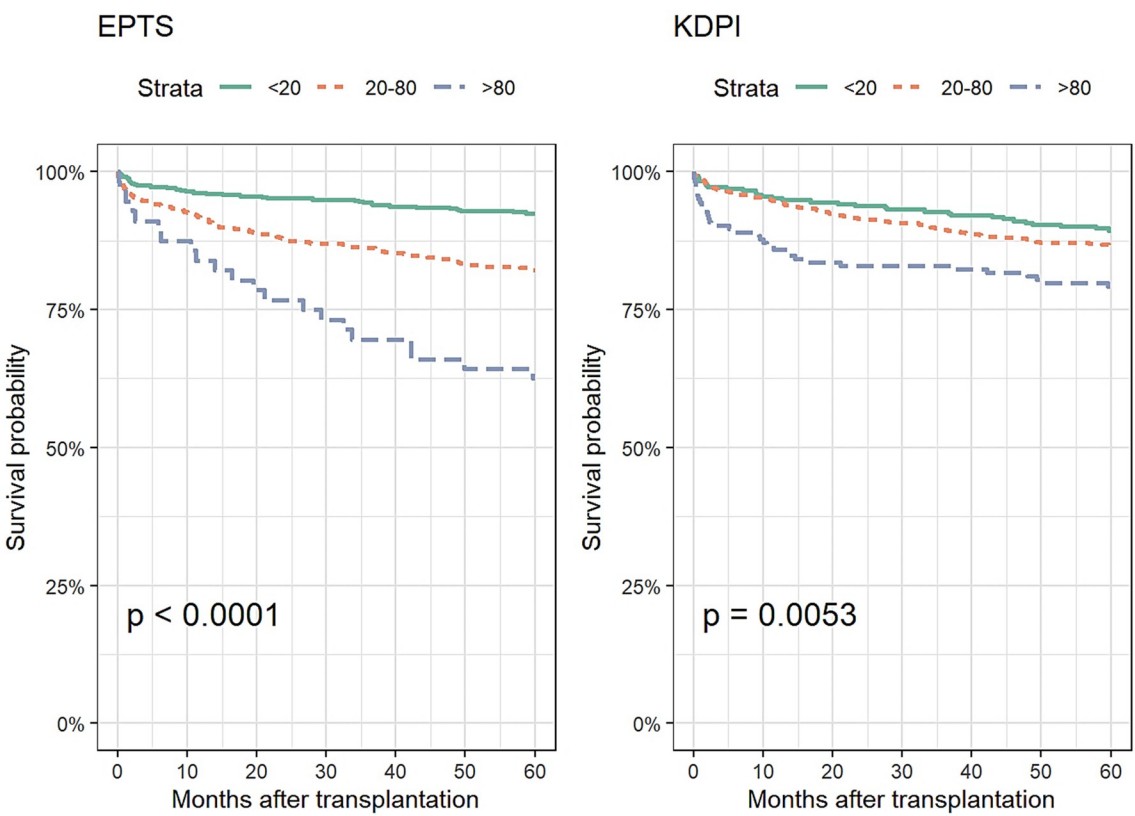

**Fig 4. Kaplan Meier survival probability estimated by EPTS and KDPI stratified by age groups.**

Our results showed that the SBE tool overestimated actual survival rates in our sample. In an unadjusted analysis, post-kidney transplant survival, KDPI, and EPTS were significantly associated with 5-year patient survival. The discriminatory performance using a C- statistic was low for the KDPI model (0.59) and moderate for the EPTS and PPS models (0.65 and 0.66, respectively). These findings may reflect the limited utility of this tool in this specific population. This is not an isolated finding, in a Spanish cohort was performed an external validation of the use of SBE [6]. Similar to our results, the SBE score did not estimate accurately the actual survival of patients with the highest and lowest scores. Furthermore, similar discrimination power was reported with a Harrell's C-statistic of 0.69, 0.57, and 0.71 for unadjusted models of SBE, KDPI, and EPTS, respectively.

KDPI and EPTS scores are relevant because they are currently used in allocating deceased donor kidneys in the United States. However, their predictive performance is not ideal like other long-term outcome prediction models in kidney transplantation (Harrell's c-statistics around 0.6) [4, 20]. In our cohort, KDPI showed the lowest predictive power among the tools. This was an expected finding, as KDPI was designed to predict graft rather than patient survival, but it has been evaluated as a potential indicator of patient survival. Several studies have analyzed the effect of KDPI on patient survival, Calvillo-Arbizu et al. [21] found limited predictive power in a Spanish cohort of 2734 kidney transplants (Harrell's c-statistics around 0.62). Similar findings were reported by Johannes-Lehner et al. [22] and Peters-Sengers et al. [23] reporting a limited discrimination ability of the Kidney Donor Risk Index for 5-year mortality.

EPTS showed moderate discrimination ability. When faced with potential recipients, the EPTS will correctly predict who will live longer 66% of the time. These results are similar to those reported by the original US data with a C-statistic of 0.69 [20]. In the same way, other authors have found similar discrimination power of EPTS [6, 20, 24].

The observed results show the predictive potential of the SBE score in this Colombian population. However, these data must be interpreted with caution because there are several relevant demographic differences between the US and Colombia. For instance, the cohort used to develop SBE was much more ethnically diverse, whereas our sample was constituted mainly of Hispanics. In addition, our cohort was younger and had less prevalence of diabetes, therefore, lower EPTS scores compared to North American populations [7]. We consider that these differences in predictions and survival may be caused by characteristics not considered by the scores, which include the health system, sociodemographic, clinical, and procedure characteristics of the donor and the recipient. Moreover, the Cox model revealed KDPI, EPTS, and SBE scores as independent death factors. However, the individual's prediction utility was limited, as the C-statistic provided only fair predictive accuracy, probably because these scores only consider pretransplant variables.

This study has its limitations: first, it is a single-center observational study, and even though follow-up is complete and long, it implies the weaknesses of a retrospective analysis. Second, confounding due to unmeasured variables is still possible. Moreover, our results cannot be extrapolated to subjects of other ethnic groups. Likewise, donation after cardiac death is prohibited in Colombia, so the studied scores may perform differently.

The strengths of this study include that, to our knowledge is the first study that offers external validation of the SBE score in a Colombian population.

In conclusion, our analysis demonstrated that the SBE score overestimated actual survival rates in our sample. This tool may help assess the potential benefit associated with marginal kidney use, although its estimations may not be accurate enough for non-US patients.

## Author Contributions

**Conceptualization:** Laura Nino-Torres, Alejandro Nino-Murcia.

**Data curation:** Laura Nino-Torres, Andrea García-Lopez, Nasly Patino-Jaramillo.

**Formal analysis:** Laura Nino-Torres, Andrea García-Lopez, Fernando Giron-Luque.

**Methodology:** Andrea García-Lopez.

**Project administration:** Laura Nino-Torres, Fernando Giron-Luque, Alejandro Nino-Murcia.

**Supervision:** Fernando Giron-Luque, Alejandro Nino-Murcia.

**Validation:** Laura Nino-Torres, Andrea García-Lopez, Fernando Giron-Luque.

**Visualization:** Fernando Giron-Luque.

**Writing – original draft:** Laura Nino-Torres, Andrea García-Lopez, Nasly Patino-Jaramillo, Fernando Giron-Luque, Alejandro Nino-Murcia.

**Writing – review & editing:** Laura Nino-Torres, Andrea García-Lopez, Nasly Patino-Jaramillo, Fernando Giron-Luque, Alejandro Nino-Murcia.

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
