## [Decision Letter · Decision Letter 0]

11 Apr 2023

PONE-D-23-01716Predicting 5-year survival after kidney transplantation in Colombia using The Survival Benefit Estimator ToolPLOS ONE

Dear Dr. Nino-Murcia,

Thank you for submitting your manuscript to PLOS ONE. After careful consideration, we feel that it has merit but does not fully meet PLOS ONE’s publication criteria as it currently stands. Therefore, we invite you to submit a revised version of the manuscript that addresses the points raised during the review process.

We look forward to receiving your revised manuscript.

Kind regards,

Ali M Shendi

Academic Editor

PLOS ONE

Journal Requirements:

(1) https://uvadoc.uva.es/bitstream/handle/10324/50759/Validation-survival-benefit-estimator-tool.pdf?isAllowed=y&sequence=1

In your revision ensure you cite all your sources (including your own works), and quote or rephrase any duplicated text outside the methods section. Further consideration is dependent on these concerns being addressed.

"The authors received no specific funding for this work."

7. PLOS requires an ORCID iD for the corresponding author in Editorial Manager on papers submitted after December 6th, 2016. Please ensure that you have an ORCID iD and that it is validated in Editorial Manager. To do this, go to ‘Update my Information’ (in the upper left-hand corner of the main menu), and click on the Fetch/Validate link next to the ORCID field. This will take you to the ORCID site and allow you to create a new iD or authenticate a pre-existing iD in Editorial Manager. Please see the following video for instructions on linking an ORCID iD to your Editorial Manager account: https://www.youtube.com/watch?v=_xcclfuvtxQ

8. Your ethics statement should only appear in the Methods section of your manuscript. If your ethics statement is written in any section besides the Methods, please delete it from any other section. 

9. Please include a separate caption for each figure in your manuscript.

Additional Editor Comments:

1- The 2nd and 3rd sentences (in page 3 line 64) are repeat. Please omit "Organ shortages and limited resources makes important to improve graft and patient survival."

2- In page 3, line 76: the sentence "A tool developed by Bae S. et al. (Survival Benefit Estimator, SBE) helps estimate post-transplant survival based on donor and recipient condition using the KDPI and EPTs scores." needs rephrasing....

I suggest "Survival Benefit Estimator (SBE) is a tool developed by Bae S. et al. to help estimate post-transplant survival based on donor and recipient characteristics using the KDPI and EPTs scores".

3- The hyperlink in page 5 line 117 works only after deleting (http://) to be "www.transplantmodels.com/kdpi-epts/"

4-The ethical statement and consent are to be mentioned under the methods section only.

In this regard, I think the relevant consent here is the study participation, which was considered in page 12 line 254, rather than the donation consent in page 6 line 139! If I understood well. Please revise.

5- Some data which are not normally distributed were represented by mean (SD), e.g. Dialysis time and EPTS in table 1. I recommend revising data representation using a test of normality.

6- Revise figure orders in light of their appearance in the text (Figure 2 & figure 3)

Reviewers' comments:

Reviewer's Responses to Questions

**Comments to the Author**

1. Is the manuscript technically sound, and do the data support the conclusions?

Reviewer #1: Yes

Reviewer #2: Partly

2. Has the statistical analysis been performed appropriately and rigorously? 

Reviewer #1: Yes

Reviewer #2: No

3. Have the authors made all data underlying the findings in their manuscript fully available?

Reviewer #1: Yes

Reviewer #2: Yes

4. Is the manuscript presented in an intelligible fashion and written in standard English?

Reviewer #1: Yes

Reviewer #2: Yes

5. Review Comments to the Author

Reviewer #1: I reviewed the manuscript PONE-D-23-01716 entitled " Predicting 5-year survival after kidney transplantation in Colombia using The Survival Benefit Estimator Tool" by Nino-Torres et al.

The paper describes the use of the Survival benefit estimator -SBE- that uses ETPS/KDPI tools in order to predict 5-year patient survival in a cohort of kidney transplant patients from 4 columbian centers. These tools have been generated in kidney transplant candidates/recipients from the US and it is therefore important to know if they are transposable in an another setting with a population of a different genetic background.

The study is well-designed and well-conducted : it included 1145 KTx recipients; of those 814 were recipients of deceased donors. As compared to stay on the waiting list the survival benefit of beeing kidney-transplanted was 18.7% at 5-years, and even higher when the recipient was older than the age of 60y.

However, in conclusion the authors state that their " analysis demonstrated that SBE score underestimated actual survival rates in their sample. This tool may be helpful to assess the potential benefit associated to marginal kidney use although its estimations may not be accurate enough for non-US patients".

I think that the paper is more suited for a specific journal dedicated to Transplantation.

Reviewer #2: - Please show the flow chart of patients enrolled in the study. The author excluded patients with insufficient information and patients who had follow up times of less than 5 years. To demonstrate numbers of these excluded patients should strengthen your study performance.

- If possible, mortality outcome should be traced through the Civil Registration for the completeness. If you have already done it, please describe in your paper.

- The main cause of CKD was unknown according to Table 1, not glomerular as in the line 146.

- Distribution of KDPI and EPTS might be shown in quintile as same as PPS. This might reveal spectrum of differences in each KDPI/EPTS categories between your population and the US.

- Adjused Cox proportional hazard model is needed for complete statistical analysis. Please performed adjusted model for significant recipient characteristic which can affect survival outcome. For PPS, I propose for adjustment for both donor and recipient factors.

- Please check the Figure 3. The estimated survival of patients in the first quintile is too low. When calculated with KDPI-EPTS Survival Benefit Estimator, the 5-year survival rate of 58% is found in patients with EPTS 95% received kidney from donor KDPI of 95%. However, your cohort had small proportion of EPTS/KDPI > 80% (< 20% of all patients/donors).

- The table 3: the mean predictive kidney transplant survival is better than actual patient survival. But the Figure 3: Actual survival is better than estimated survival in every quintiles. Why are the results not in the same direction? With the result in Figure 3, the mean actual survival should be higher than that shown in Table 3.

- Please discuss regarding higher mortality rate in your population, compared to US data in SBE study. Enrolled patients in your cohort are younger and have less prevalence of DM and also less mean EPTS than SBE study. The Columbian donor are also better than US population; less mean KDPI.

- The C-stat of these pretransplantation model is not relatively high (around 0.6 - 0.7) because they included only pretransplant factors, but not post transplant characteristics such as immunosuppression, adherence, etc. These might be included in the discussion part.

- In conclusion, as I ask regarding the Table 3 and Figure 3. If predicted survival is better than actual survival, the phrase "SBE score underestimated actual survival rates in our sample" (Line 242) might not be correct. But with inferior transplant system and post transplant care might result in inferior transplant outcome.

6. PLOS authors have the option to publish the peer review history of their article (what does this mean?). If published, this will include your full peer review and any attached files.

Reviewer #1: No

Reviewer #2: No

---

## [Author Response · Author response to Decision Letter 0]

23 May 2023

Editor comments

1. The 2nd and 3rd sentences (in page 3 line 64) are repeat. Please omit "Organ shortages and limited resources makes important to improve graft and patient survival."

R/ We have deleted the sentence: "Organ shortages and limited resources makes important to improve graft and patient survival."

2. In page 3, line 76: the sentence "A tool developed by Bae S. et al. (Survival Benefit Estimator, SBE) helps estimate post- transplant survival based on donor and recipient condition using the KDPI and EPTs scores." needs rephrasing....

R/ We have rewritten the sentence with your kind suggestion " Survival Benefit Estimator (SBE) is a tool developed by Bae S. et al. to help estimate post-transplant survival based on donor and recipient characteristics using the KDPI and EPTs scores."

3. The hyperlink in page 5 line 117 works only after deleting (http://) to be www.transplantmodels.com/kdpi-epts/

R/ We have changed the hyperlink to “www.transplantmodels.com/kdpi-epts/”

4. The ethical statement and consent are to be mentioned under the methods section only. In this regard, I think the relevant consent here is the study participation, which was considered in page 12 line 254, rather than the donation consent in page 6 line 139! If I understood well. Please revise.

R/ Our ethics statement and consent were changed and only written in the methods section. And we have included the following “Since this retrospective cohort was considered a non-risk study by the Colombian Resolution 8430 of 1993 (19), the ethics committee waived the requirement of informed consent.”

5. Some data which are not normally distributed were represented by mean (SD), e.g. Dialysis time and EPTS in table 1. I recommend revising data representation using a test of normality.

R/ We have performed normality tests for all quantitative variables (Kolmogorov Smirnov, Shapiro Wilk, and histograms), and no variable had a normality distribution. Therefore, we have adjusted the text and tables into the median and interquartile range. 

6. Revise figure orders in light of their appearance in the text (Figure 2 & Figure 3)

R/ We have reordered the figures by their appearance in the text.

Reviewer 1 comments

The paper describes the use of the Survival benefit estimator -SBE- that uses ETPS/KDPI tools in order to predict 5-year patient survival in a cohort of kidney transplant patients from 4 columbian centers. These tools have been generated in kidney transplant candidates/recipients from the US and it is therefore important to know if they are transposable in an another setting with a population of a different genetic background. The study is well-designed and well-conducted : it included 1145 KTx recipients; of those 814 were recipients of deceased donors. As compared to stay on the waiting list the survival benefit of beeing kidney-transplanted was 18.7% at 5-years, and even higher when the recipient was older than the age of 60y. However, in conclusion the authors state that their " analysis demonstrated that SBE score underestimated actual survival rates in their sample. This tool may be helpful to assess the potential benefit associated to marginal kidney use although its estimations may not be accurate enough for non-US patients".

I think that the paper is more suited for a specific journal dedicated to Transplantation.

R / Dear reviewer 1, we sincerely thank you for your comments and revision. We consider that this article proposes an interesting topic for the wide medical community since these tools are used worldwide and not always validated in local populations. Once more, thank you. 

Reviewer 2

1. Please show the flow chart of patients enrolled in the study. The author excluded patients with insufficient information and patients who had follow up times of less than 5 years. To demonstrate numbers of these excluded patients should strengthen your study performance.

R/ We have included a chart of the inclusion an exclusion (figure 1)

2. If possible, mortality outcome should be traced through the Civil Registration for the completeness. If you have already done it, please describe in your paper.

R/ We have a strict patient follow-up. Even when they miss appointments, we are constantly calling them and their families and reviewing the ADRES database that gives information about the patient, including mortality information. We have included this information on page 4. 

3. The main cause of CKD was unknown according to Table 1, not glomerular as in the line 146.

R/ We have fixed this sentence: "The main known cause of CKD was glomerular (n=209;18.3%).”

4. Distribution of KDPI and EPTS might be shown in quintile as same as PPS. This might reveal spectrum of differences in each KDPI/EPTS categories between your population and the US.

R/ We have adjusted the figure and included KDPI and EPTS scores (Figure 3).

5. Adjusted Cox proportional hazard model is needed for complete statistical analysis. Please performed adjusted model for significant recipient characteristic which can affect survival outcome. For PPS, I propose for adjustment for both donor and recipient factors.

R/ We have made the Adjusted Cox proportional hazard model for PPS, EPTS, and KDPI. In the statistical analysis, we included “We used an unadjusted and adjusted Cox proportional hazard regression to obtain Hazard ratios (HRs) with 95% confidence interval (CI) of three models (i) PPS, (ii) KDPI and (iii) EPTS. Adjusted variables were selected by clinical and statistical significance in the bivariate analysis, was excluded the variables that are considered in each score (PPS, KDPI, EPTS)” (Page 5). We described what variables were considered in the model “Adjusted HR for KDPI was adjusted with recipients age, sex, diabetes history, months in dialysis and CKD etiology. The EPTS model was adjusted by donors' age, sex, BMI, death cause, diabetes, and hypertension history. Finally, the predicted survival model was adjusted by recipients’ sex, diabetes history, months in dialysis, CKD etiology, donors' sex, BMI, and cause of death” (Page 9). Finally, we presented the results in Table 4. 

6. Please check the Figure 3. The estimated survival of patients in the first quintile is too low. When calculated with KDPI-EPTS Survival Benefit Estimator, the 5-year survival rate of 58% is found in patients with EPTS 95% received kidney from donor KDPI of 95%. However, your cohort had small proportion of EPTS/KDPI > 80% (< 20% of all patients/donors).

R/ We have adjusted the figure and values with the predicted and actual survival divided in quintiles by SBE results. The quintiles were distributed to create five similar groups between the highest and lowest prediction (Figure 3)

7. The table 3: the mean predictive kidney transplant survival is better than actual patient survival. But the Figure 3: Actual survival is better than estimated survival in every quintiles. Why are the results not in the same direction? With the result in Figure 3, the mean actual survival should be higher than that shown in Table 3.

R/ We have corrected Figure 3 into adequate quintiles. Therefore, the information in Table 3 and Figure 3 is in the same direction. 

8. Please discuss regarding higher mortality rate in your population, compared to US data in SBE study. Enrolled patients in your cohort are younger and have less prevalence of DM and also less mean EPTS than SBE study. The Columbian donor are also better than US population; less mean KDPI.

R/ We have included the following “We consider that these differences in predictions and survival may be caused by characteristics not considered by the scores, which include the health system, sociodemographic, clinical, and procedure characteristics of the donor and the recipient.”

9. The C-stat of these pretransplantation model is not relatively high (around 0.6 - 0.7) because they included only pretransplant factors, but not post transplant characteristics such as immunosuppression, adherence, etc. These might be included in the discussion part.

R/ We have included the following “However, the utility of prediction for the individual was limited, as the C-statistic provided only fair predictive accuracy, probably because this scores only consider pretransplant variables.”

10. In conclusion, as I ask regarding the Table 3 and Figure 3. If predicted survival is better than actual survival, the phrase "SBE score underestimated actual survival rates in our sample" (Line 242) might not be correct. But with inferior transplant system and post transplant care might result in inferior transplant outcome.

R/ We have corrected the discussion and conclusions as SBE scores overestimated the actual survival of our patients. 

Dear Reviewer 2, thank you for the feedback and comments. We expect the manuscript to improve with our changes due to your recommendations.

---

## [Decision Letter · Decision Letter 1]

3 Aug 2023

Predicting 5-year survival after kidney transplantation in Colombia using The Survival Benefit Estimator Tool

PONE-D-23-01716R1

Dear Dr. Nino-Murcia,

We’re pleased to inform you that your manuscript has been judged scientifically suitable for publication and will be formally accepted for publication once it meets all outstanding technical requirements.

Kind regards,

Academic Editor

PLOS ONE

Additional Editor Comments (optional):

Reviewers' comments:

Reviewer's Responses to Questions

**Comments to the Author**

1. If the authors have adequately addressed your comments raised in a previous round of review and you feel that this manuscript is now acceptable for publication, you may indicate that here to bypass the “Comments to the Author” section, enter your conflict of interest statement in the “Confidential to Editor” section, and submit your "Accept" recommendation.

Reviewer #1: All comments have been addressed

Reviewer #2: All comments have been addressed

2. Is the manuscript technically sound, and do the data support the conclusions?

Reviewer #1: Yes

Reviewer #2: Yes

3. Has the statistical analysis been performed appropriately and rigorously? 

Reviewer #1: Yes

Reviewer #2: Yes

4. Have the authors made all data underlying the findings in their manuscript fully available?

Reviewer #1: Yes

Reviewer #2: Yes

5. Is the manuscript presented in an intelligible fashion and written in standard English?

Reviewer #1: Yes

Reviewer #2: Yes

6. Review Comments to the Author

Reviewer #1: This is a very interesting paper authors that will be valuable for the transplant /nephrology community.

The authors have addressed my comments.

No additional comments

Reviewer #2: The authors respond to all of my comments in the revision part. Your study might improve transplantation system in your country.

7. PLOS authors have the option to publish the peer review history of their article (what does this mean?). If published, this will include your full peer review and any attached files.

Reviewer #1: No

Reviewer #2: **Yes: **Nuttasith Larpparisuth

---

## [Editor Report · Acceptance letter]

16 Aug 2023

PONE-D-23-01716R1 

Predicting 5-year survival after kidney transplantation in Colombia using The Survival Benefit Estimator Tool 

Dear Dr. Nino-Murcia:

I'm pleased to inform you that your manuscript has been deemed suitable for publication in PLOS ONE. Congratulations! Your manuscript is now with our production department. 

Kind regards, 

on behalf of

Dr. Robert Jeenchen Chen 

Academic Editor

PLOS ONE